# Hierarchical Poset Decoding for Compositional Generalization in Language

**Yinuo Guo**[1]*, **Zeqi Lin**[2], **Jian-Guang Lou**[2], **Dongmei Zhang**[2]
[1]Key Laboratory of Computational Linguistics School of EECS,
Peking University, Beijing, China;
[2]Microsoft Research Asia, Beijing, China
[1]gyn0806@pku.edu.cn
[2]{Zeqi.Lin, jlou, dongmeiz}@microsoft.com

## Abstract

We formalize human language understanding as a structured prediction task where the output is a *partially ordered set (poset)*. Current encoder-decoder architectures do not take the poset structure of semantics into account properly, thus suffering from poor compositional generalization ability. In this paper, we propose a novel hierarchical poset decoding paradigm for compositional generalization in language. Intuitively: (1) the proposed paradigm enforces partial permutation invariance in semantics, thus avoiding overfitting to bias ordering information; (2) the hierarchical mechanism allows to capture high-level structures of posets. We evaluate our proposed decoder on *Compositional Freebase Questions (CFQ)*, a large and realistic natural language question answering dataset that is specifically designed to measure compositional generalization. Results show that it outperforms current decoders.

## 1 Introduction

Understanding semantics of natural language utterances is a fundamental problem in machine learning. Semantics is usually invariant to permute some components in it. For example, consider the natural language utterance "*Who influences Maxim Gorky and marries Siri von Essen?*". Its semantics can be represented as either "$\exists x : INFLUENCE(x, Maxim\_Gorky) \wedge MARRY(x, Siri\_von\_Essen)$" or "$\exists x : MARRY(x, Siri\_von\_Essen) \wedge INFLUENCE(x, Maxim\_Gorky)$". For a complex natural language utterance, there would be many equivalent meaning representations. However, standard neural encoder-decoder architectures do not take this partial permutation invariance into account properly: (1) sequence decoders (Figure 1(a)) maximize the likelihood of one certain meaning representation while suppressing other equivalent good alternatives (Sutskever et al., 2014; Bahdanau et al., 2014); (2) tree decoders (Figure 1(b)) predict permutation invariant components respectively, but there are still certain decoding orders among them (i.e., target trees are ordered trees) (Dong and Lapata, 2016; Polosukhin and Skidanov, 2018).

If we ignore the partial permutation invariance in semantics and predict them in a certain order using these standard neural encoder-decoder architectures, the learned models will always have poor compositional generalization ability (Keysers et al., 2020). For example, a model has trained on "*Who influences Maxim Gorky and marries Siri von Essen?*" and "*Who influences A Lesson in Love's art director*", but it cannot understand and produce the correct semantic meaning of "*Who influences A Lesson in Love's art director and marries Siri von Essen?*". Here compositional generalization means the ability to understand and produce a potentially infinite number of novel combinations of

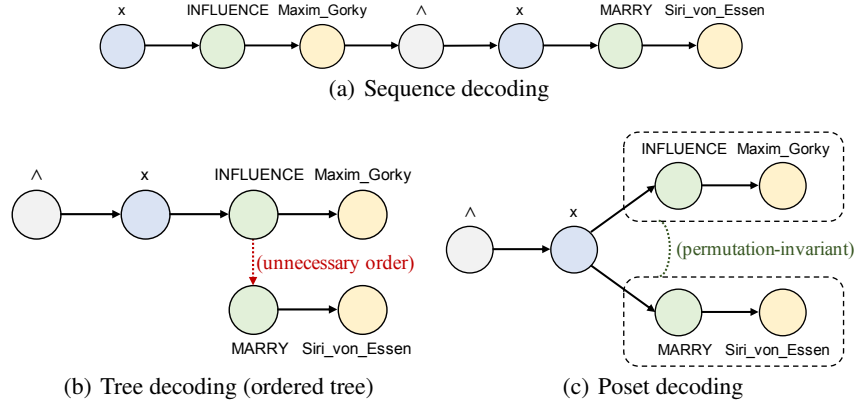

(a) Sequence decoding

(b) Tree decoding (ordered tree)     (c) Poset decoding

Figure 1: Three decoding paradigms.

known components, even though these combinations have not been previously observed (Chomsky, 1965).

The intuition behind this is: imposing additional ordering constraints increases learning complexity, which makes the learning likely to overfit to bias ordering information in training distribution. Concretely, earlier generated output tokens have a profound effect on the later generated output tokens (Mehri and Sigal, 2018; Wu et al., 2018). For example, in current sequence decoder (Figure 1(a)) and tree decoder (Figure 1(b)), "*MARRY*" is generated after "*INFLUENCE*", though these two tokens have no sequential order in actual semantics. This makes the decoding of "*MARRY*" likely to be impacted by bias information about "*INFLUENCE*" learned from training distribution (e.g.d, if the co-occurrence of "*INFLUENCE*" and "*MARRY*" has never been observed in the training data, the decoder will mistakenly choose not to predict "*MARRY*").

This problem is expected to be alleviated through properly modelling *partially-ordered set (poset)* structure in the decoding process. Figure 1(c) shows an intuitive explanation. Output token "$x$" should have two next tokens: "*INFLUENCE*" and "*MARRY*". They should be predicted from the same context in a parallel fashion, independently from each other. Partial permutation invariance in semantics should be enforced in the decoding process, thus the model can focus more on generalizable substructures rather than bias information.

In this paper, we propose a novel hierarchical poset decoding paradigm for compositional generalization in language. The basic idea for poset decoding is to decode topological traversal paths of a poset in a parallel fashion, thus preventing the decoder from learning bias ordering information among permutation invariant components. Moreover, inspired by the natural idea that semantic primitives and abstract structures should be learned separately (Russin et al., 2019; Li et al., 2019; Lake, 2019; Gordon et al., 2020), we also incorporate a hierarchical mechanism to better capture the compositionality of semantics.

We evaluate our hierarchical poset decoder on *Compositional Freebase Questions (CFQ)*, a large and realistic natural langauge question answering dataset that is specifically designed to measure compositional generalization (Keysers et al., 2020). Experimental results show that the proposed paradigm can effectively enhance the compositional generalization ability.

## 2   The CFQ Compositional Tasks

The CFQ dataset (Keysers et al., 2020) contains natural language questions (in English) paired with the meaning representations (SPARQL queries against the Freebase knowledge graph). Figure 2 shows an example instance.

To comprehensively measure a learner's compositional generalization ability, CFQ dataset is split into train and test sets based on two principles:

(1) *Minimizing primitive divergence:* all primitives present in the test set are also present in the train set, and the distribution of primitives in the train set is as similar as possible to their distribution

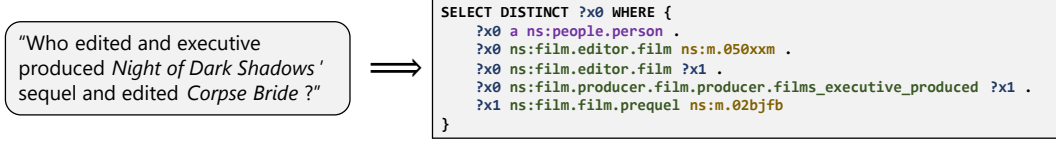

Figure 2: An instance in CFQ dataset (natural language question $\Rightarrow$ SPARQL query).

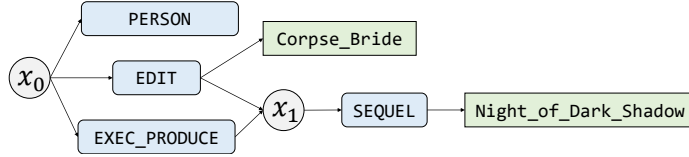

Figure 3: We model the semantic meaning of a question as a poset. Every poset can take the form of a DAG (Directed Acyclic Graph). There are three types of elements in semantic posets: variables (rounds), predicates (rectangles with rounded corners), and entities (rectangles). Each variable and entity is unique in every semantic poset.

in the test set. (2) *Maximizing compound divergence:* the distribution of compounds (i.e., logical substructures in SPARQL queries) in the train set is as different as possible from the distribution in the test set. The second principle guarantees that the task is compositionally challenging. Following these two principles, three different *maximum compound divergence* (MCD) dataset splits are constructed. Standard encoder-decoder architectures achieve an accuracy larger than 95% on a random split, but the mean accuracy on the MCD splits is below 20%.

In CFQ dataset, all SPARQL queries are controlled within the scope of *conjunctive logical queries*. Each query $q$ can be written as:

$$
\begin{aligned}
q &= \exists x_0, x_1, ..., x_m : c_1 \wedge c_2 \wedge ... \wedge c_n, \\
\text{where} \quad c_i &= (h, r, t), \quad h, t \in \{x_0, x_1, ..., x_m\} \cup E, \quad r \in P
\end{aligned}
\tag{1}
$$

In Equation 1, $x_0, x_1, ..., x_m$ are variables, $E$ is a set of entities, and $P$ is a set of predicates (i.e., relations). From the perspective of logical semantics, conjunctive clauses (i.e., $c_1, c_2, ..., c_n$) are invariant under permutation. In CFQ experiments, queries are linearized through sorting these clauses based on alphabetical order, rather than the order of appearance in natural language questions. Our hyphothesis is that this imposed ordering harms the learning of compositionality in language, so the key is to empower the decoder with the ability to enforce partial permutation invariance in semantics.

## 3  Poset

In this section, we study the task of generating a finite poset $Y$ given input $X$.

A poset $Y$ is a set of elements $\{y_1, y_2, ..., y_{|Y|}\}$ with a partial order "$\preceq$" defined on it. Let $\prec$ be the binary relation on $Y$ such that $y_i \prec y_j$ if and only if $y_i \preceq y_j$ and $y_i \neq y_j$. Let $\prec\!\!\cdot$ be the covering relation on $Y$ such that $y_i \prec\!\!\cdot y_j$ if and only if $y_i \prec y_j$ and there is no element $y_k$ such that $y_i \prec y_k \prec y_j$. Every poset can be represented as a *Hasse diagram*, which takes the form of a directed acyclic graph (DAG): $\{y_1, y_2, ..., y_{|Y|}\}$ represents a vertex set and $\prec\!\!\cdot$ represents the edge set over it. Then, we can use the language of graph theory to discuss posets.

Let $\mathcal{V}$ be the vocabulary of output tokens for poset. For each $y \in Y$, we assign an output token $\mathcal{L}_Y(y) \in \mathcal{V}$ as its label. In this paper, we focus on $\mathcal{L}_Y$ that meets the following two properties:

1. If $y_i, y_j, y_k \in Y$, $y_j \neq y_k$, $y_i \prec\!\!\cdot y_j$, and $y_i \prec\!\!\cdot y_k$, then $\mathcal{L}_Y(y_j) \neq \mathcal{L}_Y(y_k)$. Intuitively, outgoing vertices of the same vertex must have different labels.

2. There exists $\hat{\mathcal{V}} \subset \mathcal{V}$, so that for all poset $Y$ and $y \in Y, |\{y'|y' \in Y, y' \prec\!\!\cdot y\}| \leq 1$ or $\mathcal{L}_Y(y) \in \hat{\mathcal{V}} \wedge |\{y'|y' \in Y, \mathcal{L}_Y(y') = \mathcal{L}_Y(y)\}| = 1$. Intuitively, some output tokens ($\hat{\mathcal{V}}$) can

only appear once per poset, and we restrict that only these vertices can have more than one incoming edge.

We refer to posets meeting these two properties as *semantic posets*. This is a tractable subset of posets, and it is flexible to generalize to different kinds of semantics in human language.

Take conjunctive logical queries $\mathcal{Q}$ in CFQ dataset as an example. For each query $q \in \mathcal{Q}$, we model it as a semantic poset $Y_q$ using the following principles:

- The output vocabulary $\mathcal{V} = \mathcal{V}_x \cup P \cup E$, where $\mathcal{V}_x$ is the vocabulary of variables $(x_0, x_1, ...)$, $P$ is the vocabulary of predicates, and $E$ is the vocabulary of entities. We define that $\hat{\mathcal{V}} = \mathcal{V}_x \cup E$.

- Variables and entities in $q$ are represented as vertices. Their labels are themselves.

- For each variable or entity $v$ in $q$, we define the set of all clauses in $q$ as $C_v$, in which $v$ is the head object. We also denote the set of all predicates in $C_v$ as $P_v$. We create a vertex $v'_{v,p}$ for each $p \in P_v$, let $\mathcal{L}_y(v'_{v,p}) = p$, then create an edge from $v$ to $v'_{v,p}$ (i.e., $v \prec v'_{v,p}$). Let $T_{v,p}$ be the set of tail objects in clauses that $v$ is the head object and $p$ is the predicate. For each vertex in $T_{v,p}$, we create an edge from $v'_{v,p}$ to it.

Figure 3 shows an example of semantic poset. We visualize the semantic poset as a DAG: each variable is represented as a round; each predicate is presented as a rectangle; each entity is presented as a rectangle with rounded corners.

## 4 Poset Decoding (Basic Version)

In this section, we introduce a minimal viable version of poset decoding. It can be seen as a extension of sequence decoding, in which poset structure is handled in a topological traversal-based way.

In a standard sequence decoder, tokens are generated strictly in a left-to-right manner. The probability distribution of the output token at time step $t$ (denoted as $\boldsymbol{w}_t$) is represented with a softmax over all the tokens in the vocabulary (denoted as $\mathcal{V}$):

$$\boldsymbol{w}_t = softmax(\boldsymbol{W}_o \boldsymbol{h}_t) \tag{2}$$

, where $\boldsymbol{h}_t$ is the hidden state in the decoder at time step $t$, $\boldsymbol{W}_o$ is a learnable parameter matrix that is designed to produce a $|\mathcal{V}|$-dimensional logits from $\boldsymbol{h}_t$.

When the decoding object is extended from a sequence to a semantic poset $Y$, we need to extend the sequential time steps into a more flexible fashion, that is, topological traversal paths. We define that $l = (l_1, l_2, ..., l_{|l|})$ is a topological traversal path in $Y$ ($l_1, l_2, ..., l_{|l|} \in Y$), $l_1$ is a lower bound element in $Y$ (i.e., $|\{y'|y' \in Y, y' \prec l_1\}| = 0$) and $l_1 \prec l_2 \prec ... \prec l_{|l|}$. For each topological traversal path $l$ in $Y$, we generate a hidden state $\boldsymbol{h}_l$ for it, following the way that sequence decoders uses. Without loss of generality, we describe an implementation based on attention RNN decoder:

$$\boldsymbol{h}_{l,t} = RNN(\boldsymbol{h}_{l,t-1}, \boldsymbol{e}_{l,t-1}, \boldsymbol{c}_{l,t}) \tag{3}$$

In Equation 3, $\boldsymbol{h}_{l,t}$ represents the hidden state of $l_t$, and we let $\boldsymbol{h}_l = \boldsymbol{h}_{l,|l|}$. The hidden state $\boldsymbol{h}_{l,t}$ is computed using an RNN module, of which the inputs are: $\boldsymbol{h}_{l,t-1}$ (the previous hidden state), $\boldsymbol{e}_{l,t-1}$ (the embedding of $\mathcal{L}_Y(l_{t-1})$), and the context vector $\boldsymbol{c}_{l,t}$ generated by attention mechanism.

We use $\boldsymbol{h}_l$ to predict $next(l) = \{y|y \in Y, l_{|l|} \prec y\}$. This sub-task is equivalent to the prediction of $next\_label(l) = \{\mathcal{L}_Y(y)|y \in Y, l_{|l|} \prec y\}$, because there will not be any other different topological traversal paths with the same label sequence in a semantic poset. Therefore, for each $w \in \mathcal{V}$, we perform a binary classification to predict whether $w \in next\_label(l)$, of which the probability for the positive class is computed via:

$$P(+|l, w, X) = \begin{bmatrix} 1 \\ 0 \end{bmatrix} softmax(\boldsymbol{W}_w \boldsymbol{h}_l) \tag{4}$$

---

**Algorithm 1** generate_poset (the decoding process at inference time)

---

**Inputs:** model input $X$, current poset $Y$, typological traversal path $l$, output vocabulary $\mathcal{V}$ and $\hat{\mathcal{V}}$

---

1: $next\_labels := \emptyset$, $next\_vertices := \emptyset$
2: **for** $w \in \mathcal{V}$ **do**
3: &emsp; **if** $P(+|l, w, X) > 0.5$ **then**
4: &emsp;&emsp; $next\_labels$.add($w$)
5: **for** $next\_label \in next\_labels$ **do**
6: &emsp; **if** $next\_label \in \hat{\mathcal{V}}$ **then**
7: &emsp;&emsp; $v := Y.vertices$.find($\lambda x : \mathcal{L}_Y(x) = next\_label$)
8: &emsp; **else**
9: &emsp;&emsp; $v := Y.edges$.find($\lambda x : x.src\_vertex = l_{|l|} \land \mathcal{L}_Y(x.target\_vertex) = next\_label$)
10: &emsp; **if** $v == nil$ **then**
11: &emsp;&emsp; $v :=$ new Vertex($next\_label$)
12: &emsp; $next\_vertices$.add($v$), $Y.edges$.add($(l_{|l|}, v)$)
13: **for** $v \in next\_vertices$ **do**
14: &emsp; generate_poset($X, Y, l \oplus v, \mathcal{V}, \hat{\mathcal{V}}$)
15: **return** $Y$

---

, where $\boldsymbol{W}_w$ is a learnable parameter matrix designed to produce a 2-dimensional logits from $\boldsymbol{h}_l$.

At training time, we estimate $\theta$ (parameters in the whole encoder-decoder model) via:

$$\theta^* = \arg\max \prod_{(X,Y)\in\mathcal{D}_{train}} \prod_{l\in paths(Y)} \prod_{w\in\mathcal{V}} P(+|l, w, X) \tag{5}$$

At inference time, we predict the target poset via Algorithm 1, which is initialized with $Y = \emptyset$ and $l$ is a zero-length path. An intuitive explanation for Algorithm 1 is: we decode topology traversal paths in parallel; different paths are merged at variables and entities (to ensure that each variable and entity is unique in every semantic poset).

## 5 Hierarchical Poset Decoding

Inspired by the essential idea that separating the learning of syntax and semantics helps a lot to capture the compositionality in language (Russin et al., 2019; Li et al., 2019; Lake, 2019; Gordon et al., 2020), we propose to incoporate a hierarchical mechanism into the basic poset decoding paradigm. Our hierarchical poset decoding paradigm consists of three components: sketch prediction, primitive prediction, and traversal path prediction. Figure 4 shows an example of how this hierarchical poset decoding paradigm works in the CFQ task.

### 5.1 Sketch Prediction

For each semantic poset $Y$, we define that $f : \mathcal{V} \rightarrow \mathcal{A}$ is a function that maps each output token $w \in \mathcal{V}$ in the output vocabulary to an abstract token $f(w) \in \mathcal{A}$. $\mathcal{A}$ is the vocabulary of those abstract tokens, and $|\mathcal{A}| << |\mathcal{V}|$.

Specifically, in the CFQ task, we define that: each predicate $p \in P$ should be abstracted to a token "$P$", and each entity $e \in E$ should be abstracted to a token "$E$".

Then, a *sketch* $\mathcal{S}_Y$ can be extracted from $Y$ through the following steps:

1. Initialize $\mathcal{S}_Y$ as a copy of $Y$. Then, For each $v \in \mathcal{S}_Y$, let $\mathcal{L}_{\mathcal{S}_Y}(v) \leftarrow f(\mathcal{L}_{\mathcal{S}_Y}(v))$.

2. Merge two vertexes $v_i, v_j \in \mathcal{S}_Y$, if $\mathcal{L}_{\mathcal{S}_Y}(v_i) = \mathcal{L}_{\mathcal{S}_Y}(v_j)$, $\{v'|v' \in \mathcal{S}_Y, v' \prec v_i\} = \{v'|v' \in \mathcal{S}_Y, v' \prec v_j\}$ and $\{v'|v' \in \mathcal{S}_Y, v_i \prec v'\} = \{v'|v' \in \mathcal{S}_Y, v_j \prec v'\}$. Intuitively, two vertexes should be merged, if they share the same label and neighbors.

3. Repeat Step 2, until there is no any pair of vertex meets this condition.

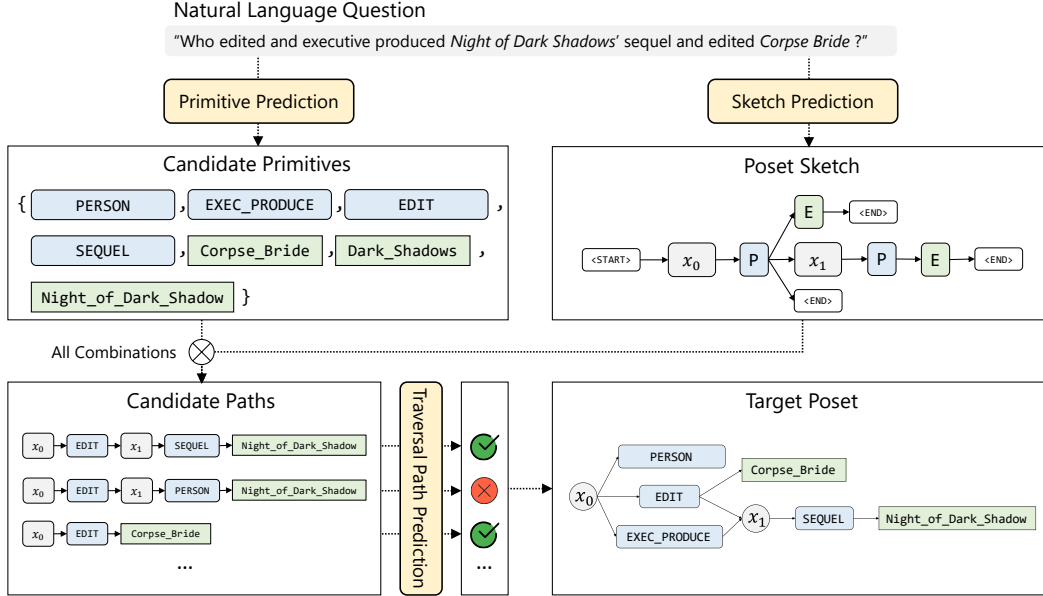

Figure 4: An example of how the hierarchical poset decoding paradigm works in the CFQ task.

The top-right corner in Figure 4 demonstrates a poset sketch as an example. This procedure garuantees that $\mathcal{S}_Y$ is still a semantic poset, with a much smaller label vocabulary $\mathcal{A}$ than $\mathcal{V}$. Therefore, given an input $X$, we can predict the sketch of its target poset via the basic poset decoding algorithm (proposed in Section 4).

## 5.2 Primitive Prediction

We define *primitives* as $prim = \{w | w \in \mathcal{V}, f(w) \neq w\}$. Given input $X$, the primitive prediction module is designed to generate a set $prim_X \subset prim$, which represents candidate primitives that are likely to be used in the target poset.

In the CFQ task, we implement the primitive prediction module by adopting phrase table induction from statistical machine translation (Koehn, 2009). More concretely, a phrase table is a collection of natural language phrases (i.e., n-grams) paired with a list of their possible translations in the target semantics. For each *(phrase, primitive)* pair, the translation probability $p(primitive|phrase)$ is inducted from the training instances $\mathcal{D}_{train}$. In our primitive prediction module, for each input question $X$, we define a primitive as a candidate primitive for $X$, if and only if there exists some phrases in $X$ such that $p(primitive|phrase)$ is larger than the threshold.

## 5.3 Traversal Path Prediction

For each topological traversal path in the predicted sketch, each vertex $v$ with $\mathcal{L}_{\mathcal{S}_Y}(v) \in \mathcal{A} \setminus \mathcal{V}$ can be viewed as a slot, which can be filled by one label in $\{w | w \in \mathcal{V}, f(w) = \mathcal{L}_{\mathcal{S}_Y}(v)\}$. We enumerate all permutations, thus obtaining a set of candidate paths. For each candidate $path$, we aim to recognize entailment relation between $path$ and the input $X$ (i.e., whether $path$ is a topological traversal path in the target poset of $X$). We implement this using an ESIM network (Chen et al., 2016):

$$P(path|X) = ESIM(X, path) \tag{6}$$

Then we deterministically re-construct the target poset from paths meeting $P(path|X) > 0.5$.

# 6 Experiments

## 6.1 Settings

**Dataset** We conduct experiments on three different splits MCD1/MCD2/MCD3 of CFQ dataset (Keysers et al., 2020). Each split contains 95k/12k/12k instances for training/development/test.

**Implementation Details** (1) We implement the sketch prediction module based on a 2-layer bidirectional GRU (Cho et al., 2014). The dimensions of the hidden state and word embedding are set to 512, 300 respectively. (2) To implement the primitive prediction module, we leverage GIZA++[2] (Och and Ney, 2003) toolkit to extract a phrase table with total 262 pairs from the train set. (3) For the traversal path prediction module, we utilize the ESIM (Chen et al., 2016) network implemented in Match-Zoo[3] (Guo et al., 2019) with the default setting. (4) The training process lasts 50 epochs with batch size 64/256 for sketch prediction and traversal path prediction, respectively. We use the Adam optimizer with default settings (in PYTORCH) and a dropout layer with the rate of 0.5.

**Evaluation Metric** We use accuracy as the evaluation metric, i.e., the percentage of the examples that are correctly parsed to their gold standard meaning representations (SPARQL queries).

## 6.2 Results

As shown in Table 1, **H**ierarchical **P**oset **D**ecoding (HPD) is superior to all baselines by a large margin on three splits. We also have the following observations:

*1. Seq2seq models have poor compositionial generalization ability.* In the first block of Table 1, we list results of Seq2seq baselines reported by Keysers et al. (2020). The mean accuracy on the MCD splits is below 20% for all these models.

*2. Tree decoder can hardly generalize to combinations that have not been previously observed in the training data.* We represent each SPARQL query as a tree and predict the structure using a tree decoder (Dong and Lapata, 2016). This baseline performs even worse than Seq2seq baselines, which suggests that: tree decoder has poor compositional generalization ability even though the compositional structures in semantics are modeled explicitly.

*3. Simplifying queries through merging similar triples helps a lot to improve the compositional generalization ability.* We speculate that it is not suitable to directly decode standard SPARQL expressions, as it breaks up the compositional structure of semantics into fine-grained triples. Therefore, we conduct experiments in which SPARQL queries are simplified through merging similar triples (the second block in Table 1). For example, given a SPARQL query consists of four triples — $(x_0, p_1, e_1)$, $(x_0, p_2, e_1)$, $(x_0, p_1, e_2)$ and $(x_0, p_2, e_2)$, we simplify it to $(x_0, (p_1, p_2), (e_1, e_2))$. Then, we use Seq2seq models to predict these simplified expressions. This improves the accuracy a lot (but still not good enough).

*4. CFQ tasks require broader compositional generalization ability beyond primitive substitution.* CGPS (Li et al., 2019) is a neural architecture specifically desgined for primitive substitution in compositional generalization. It performs poorly in CFQ tasks (4.8%/ 1.0%/1.8% accuracy on three MCD splits). Compared to CGPS, our model performs well because it can handle not only primitive substitution (through the hierarchical mechanism), but also novel combinations of bigger components (through properly modeling partial permutation invariance among these components).

## 6.3 Ablation Analysis

We conduct a series of ablation analysis to better understand how components in HPD impact overall performance (the fourth block in Table 1). Our observations are as follows:

*1. Enforcing partial permutation invariance in semantics can significantly improve compositional generalization ability.* When the poset decoding algorithm for sketch prediction is replaced with Seq2Seq/Seq2Tree model, the mean accuracy on MCD splits drops from 69.0% to 56.7%/55.9%.

Table 1: Performance of different models on three split of CFQ dataset.

| Models | MCD1 | MCD2 | MCD3 |
|---|---|---|---|
| LSTM+Attention (Keysers et al., 2020) | 28.9% | 5.0% | 10.8% |
| Transformer (Keysers et al., 2020) | 34.9% | 8.2% | 10.6% |
| Universal Transformer (Keysers et al., 2020) | 37.4% | 8.1% | 11.3% |
| LSTM+Attention (with simplified SPARQL expression) | 42.2% | 14.5% | 21.5% |
| Transformer (with simplified SPARQL expression) | 53.0% | 19.5% | 21.6% |
| Seq2Tree (Dong and Lapata, 2016) | 24.3% | 4.1% | 6.3% |
| CGPS (Li et al., 2019) | 4.81% | 1.04% | 1.82% |
| **Hierarchical Poset Decoding** | **79.6%** | **59.6%** | **67.8%** |
| with Seq2Seq-based sketch prediction | 74.3% | 45.7% | 50.2% |
| with Seq2Tree-based sketch prediction | 75.7% | 40.9% | 51.1% |
| w/o Hierarchical Mechanism | 21.3% | 6.4% | 10.1% |

Table 2: Performance of Different Components in **H**ierarchical **P**oset **D**ecoding Model.

| Split | **S**ketch Prediction (SP) | **P**rimitive **P**rediction (PP) | | | **T**raversal **P**ath Prediction (TPP) | | |
|---|---|---|---|---|---|---|---|
| | Accuracy | Precision | Recall | F1 Score | Precision | Recall | F1 Score |
| MCD1 | 91.3% | 34.4% | 99.9% | 51.1% | 96.7% | 98.9% | 97.7% |
| MCD2 | 75.1% | 35.0% | 99.9% | 51.9% | 89.2% | 87.5% | 88.4% |
| MCD3 | 74.8% | 34.1% | 99.9% | 50.8% | 92.8% | 90.8% | 91.8% |

*2. The hierarchical mechanism is essential for poset decoding paradigm.* The model without hierarchical mechanism performs poorly: dropping 58.3%/53.2%/57.7% accuracy on MCD1/MCD2/MCD3, respectively. The reason is: when the output vocabulary $\mathcal{V}$ has a large size, it's difficult to well-train the basic poset decoding algorithm proposed in Section 4, since the classification tasks described in Equation 4 will suffer from the class imbalance problem (for each $w \in \mathcal{V}$, there are much more negative samples than positive samples.) The hierarchical mechanism alleviates this problem (because $|\mathcal{A}| << |\mathcal{V}|$), thus bringing large profits.

The ablation analysis shows that both the poset decoding paradigm and the hierarchical mechanism are of high importance for capturing the compositionality in language.

### 6.4 Break-Down Analysis

To understand the source of errors, we conduct evaluation for different components (i.e., sketch prediction, primitive prediction, and traversal path prediction) in our model. Table 2 shows the results.

**Sketch Prediction** For 8.7%/24.9%/25.2% of test cases in MCD1/MCD2/MCD3, our model fails to predict their correct poset sketches. Sketch is an abstraction of high-level structural information in semantics, corresponding to syntactic structures of natural language utterances. We guess the reason that causes these error cases is: though our decoder captures the compositionality of semantics, the encoder deals with input utterances as sequences, thus implicit syntactic structures of inputs are not well captured. Therefore, the compositional generalization ability is expected to be further improved through enhancing the encoder with the compositionality (Dyer et al., 2016; Kim et al., 2019; Shen et al., 2018) and build proper attention mechanism between the encoder and our hierarchical poset decoder. We leave this in future work.

**Primitive Prediction** Our primitive prediction module performs high recall (99.9%/99.9%/99.9%) but relative low precision (34.4%/35.0%/34.1%). Currently, our primitive prediction module is based on a simple phrase table. In future work, we plan to improve the recall through leveraging contextual information in utterances, thus alleviating the burden of traversal path prediction.

**Traversal Path Prediction** Our traversal path prediction module performs well on classifying candidate paths. This is consistent with the finding that neural networks exhibit compositional

generalization ability on entailment relation reasoning (Mul and Zuidema, 2019). However, for 11.7%/15.5%/7.0% of test cases in MCD1/MCD2/MCD3, our model fails to predict the exact collections of traversal paths. This is mainly due to the large number of candidate paths: on average, each test utterance has 6.1 positive candidate paths and 18.5 negative candidate paths. In future work, this problem is expected to be addressed through reducing the number of candidate primitives.

## 7 Related Work

**From Set Prediction to Poset Prediction**    Many kinds of data are naturally represented as a set in which elements are invariant under permutations, such as points in a point cloud (Achlioptas et al., 2017; Fan et al., 2017), objects in an image (object detection) (Rezatofighi et al., 2018; Balles and Fischbacher, 2019), nodes in a molecular graph (molecular generation) (Balles and Fischbacher, 2019; Simonovsky and Komodakis, 2018). Many researches (Vinyals et al., 2015; Rezatofighi et al., 2018; Zhang et al., 2019) focus on set prediction and explore to take the disorder natural of set into account. In contrast, less attention has been paid to partially ordered set (or poset), in which some elements are invariant under permutation, but some are not. Poset is a widespread structure, especially in semantics of human language. In this work, we explore to properly take the poset structure of semantics into account in semantic parsing tasks (i.e., translating human language utterances to machine-interpretable semantic representations).

**Structured Prediction in Semantic Parsing**    In recent work on developing semantic parsing systems, various methods have been proposed based on neural encoder-decoder architectures. Semantic meanings can be expressed as different data structures, such as sequence (Sutskever et al., 2014; Bahdanau et al., 2014), tree (Dong and Lapata, 2016; Polosukhin and Skidanov, 2018), and graph (Buys and Blunsom, 2017; Damonte et al., 2017; Lyu and Titov, 2018; Fancellu et al., 2019), then different decoders are designed to predict them. Our work incoporates the partial permutation invariance of semantics into the decoding process, thus significantly improving the compositional generalization ability. Additionally, SQLNet (Xu et al., 2017) and Seq2SQL (Zhong et al., 2017) are two semantic parsing systems that also emphasize the importance of partial permutation invariance: SQLNet is specifically designed for permutation invariant WHERE clauses in SQL; Seq2SQL proposes to leverage reinforcement learning to better generate partial ordered parts in semantics. Comparing to them, our work can generalize to various semantic formalisms that have the form of poset, and it will not suffer from the sparse reward problem of reinforcement learning.

**Compositional Generalization**    Compositionality is the ability to interpret and generate a possibly infinite number of constructions from known constituents, and is commonly understood as one of the fundamental aspects of human learning and reasoning (Chomsky, 1965; Minsky, 1988). In contrast, modern neural networks struggle to capture this important property (Lake and Baroni, 2018; Loula et al., 2018). Recently, various efforts have been made to enable human-level compositional ability in neural networks. Russin et al. (2019) and Li et al. (2019) focus on separating the learning of syntax and semantics. Lake and Baroni (2018) introduces a meta Seq2seq approach for learning the new primitives. Gordon et al. (2020) formalizes the type of compositionality skills as equivariance to a certain group of permutations. These methods make successful on primitive tasks (Lake and Baroni, 2018), which is the simplest part of compositionality. In this work, we aim to obtain broader compositional generalization ability with a poset decoding paradigm and a hierarchical mechanism, which shows promising results on CFQ (Keysers et al., 2020) dataset.

## 8 Conclusions

In this paper, we propose a novel hierarchical poset decoding paradigm for compositional generalization in language. The main idea behind this is to take the poset structure into account, thus avoiding overfitting to bias ordering information. Moreover, we incorporate a hierarchical mechanism to better capture the compositionality of semantics. The competitive performances on CFQ tasks demonstrate the high compositional generalization ability of our proposed paradigm. In future work, we expect to extend our proposed paradigm to tackle more poset prediction tasks. We also hope that this work can help ease future research towards compositional generalization in machine learning.

## Acknowledgments and Disclosure of Funding

We thank the anonymous reviewers for their valuable comments. Specially, we thank Prof. Junfeng Hu for his guidance and support during the first author's graduate school career.

## Broader Impact

In this paper, authors introduce a hierarchical poset decoding paradigm to improve compositional generalization ability of neural encoder-decoder architectures for natural language understanding.

Poset structure is important in not only natural language understanding, but also many other areas such as chemistry (e.g., molecular structure), computer vision (e.g., scene graph) and software engineering (e.g., software dependencies). Our work explores to better predict poset structures, thus it may have implicit positive impact to these areas.

Compositional generalization is a fundamental property of human intelligence, and it is essential to equip machine learning systems with this ability. This could lead to positive societal implications: (1) it could help machine learning systems to avoid being affected by biases during training data collection process; (2) it could help machine learning systems reduce their reliance on massive amounts data, thus improving intelligence while saving resources.

In this paper, we focus on a natural language question answering task. We consider that there will not be certain societal consequences nor ethical aspects in the near future.

## Footnotes

*Work done during an internship at Microsoft Research.

[2]https://github.com/moses-smt/giza-pp.git

[3]https://github.com/NTMC-Community/MatchZoo-py

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
