[Reviews · NeurIPS 2020]

Review 1

Summary and Contributions: This paper presents two mechanisms for enhancing the decoder of an encoder-decoder architecture to achieve better compositional generalization in language understanding. The decoder includes three parts: sketch prediction, primitive prediction, and traversal path prediction. The authors evaluates the proposed method on a large natural language QA dataset (CFQ) for measuring compositional generalization. One contribution is that it enforces permutation invariance among partial order sets(poset) by decoding the topological traversal paths of a poset in parallel. Another one is hierarchical mechanism for capturing high-level structure of poset. Ablation study demonstrates that both poset and hierarchical decoding are important for extracting compositionality in language.

Strengths: + The hierarchical poset decoding paradigm is novel and interesting. It makes better compositional generalization than both sequence and tree decoding paradigms. + By enforcing partially order permutation invariance, it avoids overfitting to strict ordering/position information. + Hierarchical decoding alleviates class imbalance problem. + The paper provides detailed error analysis which helps readers to better understand the proposed method and to get some ideas on how to further improve the method.

Weaknesses: The paper focuses on enhancing decoder part with compositional generalization (CG). However, in order to achieve full CG, the encoder part also needs to be equipped with CG.

Correctness: The claims and empirical methodology are reasonable.

Clarity: The paper is well written.

Relation to Prior Work: The paper discussed how this work differs from previous contributions.

Reproducibility: Yes

Additional Feedback: P6L199: "Simplifying queries through merging similar triples helps a lot to improve the compositional generalization ability". My understanding is that query simplification is done manually. It is better to do it automatically. [After rebuttal]: After reading the authors feedback and other reviews' comments, I agree that the authors need to improve the clarity of the presentation by providing concrete running examples as shown in their rebuttal. The paper would be stronger if it evaluates on more datasets. However, I do think both poset decoding and hierarchical mechanism are in the right direction to achieve compositional generalization. Getting the last bit of performance requires this complexity.


Review 2

Summary and Contributions: This paper proposes a new architecture based on a few important components in order to improve performance on compositional generalization tasks in language. Instead of a standard seq2seq architecture, the authors propose a "seq2poset" setup, where the decoder in a seq2seq architecture outputs partial ordered sets instead of sequences or trees. This fits well with semantic parsing tasks (in this case, text to SPARQL queries), in which clauses often do not require a strict order. Another important component of the approach is the use of a "sketch-based" representation of the output, which is a popular approach in semantic parsing in general. The output graph corresponding to the SPARQL query is first abstracted into a more course-grained form, after which the target poset is computed using the seq2poset architecture. In order to obtain the output from the course-grained graph, a primitive prediction module is used, which provides a list of primitives that likely will be used in the output, given the input question. Given the coarse-grained representation of the input and the list of output primitive, one can now enumerate all possible permutations in which the slots can be filled exhaustively. An ESIM network is then used to recognize which paths are topological traversal paths in the target poset of the input query. Using this setup, the authors obtain new SOTA results on the CFQ dataset, outperforming other approaches by a large margin.

Strengths: The theoretical grounding of the approach is solid. The claim are sound. The empirical evaluation is good, with a few ablation analyses and a break-down of the failures. The approach is innovative, combining various different approach in an innovative way. The empirical results are impressive. The topic of compositional generalization is of interest to the NeurIPS community. UPDATE: After having read the rebutta

Weaknesses: The architecture is quite involved and the authors do not always explain it clearly. Especially Section 3 is quite abstract. This makes it difficult to understand what is going on. == After rebuttal == I am quite satisfied with the author's rebuttal. and I think the example they provide is clarifying.

Correctness: The claim, method, and empirical methodology all seem correct to me.

Clarity: Yes.

Relation to Prior Work: Yes.

Reproducibility: Yes

Additional Feedback: Section 3 is quite difficult to follow, and it would help a lot if an example was given here. Just showing some example of a SPARQL query and then the poset representation in a DAG would help a lot. A few details: - For each topological traversal path, a hidden state is created. Do we already know up front how many paths there are going to be, or is this set to some max value? - "for it" --> remove - "sequence decoders uses" --> use (it is a bit strangely phrased) - by attention mechanism --> by the attention mechanism - p.4 line 133, comma in the wrong place - Please provide some explanation to Algorithm 1. - p.5 line 148: I think there is a typo. LSY(v)_i=LSY(v)_i seems wrong. Explaining this step in a sentence or two would be useful as well.


Review 3

Summary and Contributions: This paper deals with predicting a semantic representation of language. The authors propose a model that respects the fact that semantic representations may be represented as a poset, and thus have invariance to swapping the order of elements when generating. The authors propose a three-part system, and show that it improves performance on a language to SPARQL query task compared to baseline methods. === Update I've raised my review by one point due to the strong results on CFQ; however the presentation, complexity, and the fact that it's only tested on CFQ still seem like drawbacks to me.

Strengths: Empirical evaluation: - Strong results on the CFQ dataset compared to LSTM and Transformers Significance / novelty: - The hypothesis that partial permutation invariance can improve compositional generalization is interesting

Weaknesses: Significance and Relevance: - The proposed method relies on annotated structural language representations (e.g. a SPARQL query), so its impact may be limited, e.g. to those interested in the CFQ task. Discussing other applications (or better, providing results) could help with this. - I am unclear on what the general takeaway on compositional generalization is from this work. In the introduction, the authors motivated the work as improving generalization via respecting the partial permutation invariance in a poset. However, the "w/o Hierarchical Mechanism" results in Table 2 suggest this is not the case. Is there a general take-away regarding the necessity of hierarchy? How does the hierarchical mechanism "better capture the compositionality of semantics"? Clarity of presentation: - The presentation was difficult to follow. This might be due to introducing a large number of abstractions, e.g. vertex, edge, variable, entity, label, tokens, semantic posets, head/tail objects, predicate, sketch, primitive, path. It may be helpful to reduce the number of unique terms to a minimum, and clearly connect each with the concrete running example (text to SPARQL queries).

Correctness: I did not notice any methodological errors.

Clarity: The method was difficult to understand, in particular sections 3 and 5.1-5.3. It would help to talk through a concrete example.

Relation to Prior Work: Yes

Reproducibility: Yes

Additional Feedback:


Review 4

Summary and Contributions: This paper proposes the hierarchical poset decoding for CFQ dataset. CFQ is a natural language to SparQL translation benchmark that emphasizes compositional generalization. Specifically, the primitive distributions are similar for training and test sets, but the distributions of compositional rules are very different. The natural language texts are generated using pre-defined templates, and the translation rules are manually written, thus this dataset is relatively clean compared to other semantic parsing benchmarks. Existing work shows that while standard seq2seq models could achieve >95% when training and test samples come from similar distributions, the accuracy drops to <20% when the distribution of compositional rules are very different. Instead of decoding a SparQL query as a sequence, they propose to decode the SparQL clauses as a poset, since changing the order of different SparQL clauses does not change the meaning of the SparQL query. In their full pipeline, they first predict the primitives appeared in the SparQL query, by learning a phrase table from the training set with statistical machine translation. Afterwards, they predict a poset sketch representing the SparQL, where each token could either be: (1) a variable xi; (2) an abstract token representing a predicate; or (3) an abstract token representing an entity. Finally, they enumerate all possible traversal paths by filling the abstract tokens for predicates and entities with the inferred primitives, and return the final poset with the highest decoding probability. Their approach achieves an average of ~70% accuracy on MCD splits of CFQ.

Strengths: 1. Compositional generalization is an important topic, and CFQ is a challenging benchmark for evaluation. 2. I think some proposed techniques are in the right direction of improving the performance on the CFQ dataset. Specifically, if we simply view it as a sequence-to-sequence task, then it is nearly impossible to capture the order invariant property of SparQL clauses. Explicitly taking this property into consideration could make it easier for model training.

Weaknesses: First, I feel that their poset decoding framework is not really a general-purpose approach to achieve compositional generalization in language, but is kind of specialized for decoding logical forms that include a set of conjunctive/disjunctive clauses, especially for generating SparQL or SQL queries. In this space, some existing work already proposes techniques for synthesizing unordered SQL clauses, e.g., [1][2]. In particular, although [2] does not consider compositional generalization, the hierarchical decoding process in this paper shares some high-level similarities with the sketch-based query synthesis approach in [2]. This paper lacks a discussion of related work for natural language to SQL synthesis. Second, though the results on CFQ are impressive, some important details are missing. From the original CFQ paper: "the accuracy metric requires the model response and the golden (correct) answer to be exactly equal to each other. Thus, a SPARQL query with the same clauses as the golden answer but in a different order or with some of the clauses appearing multiple times is also considered to be an error despite being equivalent to the golden answer in its meaning.". However, since HPD in this paper doesn't directly generate the SparQL query as a token sequence, there should be some alternative way to measure the equivalence, e.g., perhaps the authors consider the predictions of their approach to be correct when the set of clauses matches the ground truth, though the orders are not necessarily the same. If this is the case, then directly copying the same numbers from the CFQ paper is not appropriate, because the metric in this paper is more relaxed. Meanwhile, it is important to make sure that for all seq2seq and seq2tree baselines evaluated in this paper, the predictions are also considered as correct when the only difference is the order of predicted clauses. Also, do the authors enforce any type constraints to ensure the syntactic correctness of the predicted SparQL queries? Specifically, the poset decoding process utilizes the domain knowledge of SparQL query structures, and whether a token is a predicate or an entity, while the baselines do not seem to incorporate such information. Third, it seems that the sketch could only be transformed into SparQL clauses. How does the model generate other parts of SparQL queries? For example: 1. In Figure 2, how does the model generate "SELECT DISTINCT ?X0 WHERE"? 2. In the CFQ benchmark, there are some clauses that include the FILTER operation, e.g., FILTER (?x0 != ?x1) . FILTER (?x0 != M0). How does the model handle such cases? Lastly, for primitive prediction, what is the benefit of learning a phrase table, instead of learning a separate neural network for prediction? Note that a phrase table may work fine for this dataset since this dataset is constructed with templates, but in general it may not work so well. The low precision of primitive prediction suggests that there should be a better approach. [1] Zhong et al., Seq2SQL: Generating Structured Queries from Natural Language using Reinforcement Learning. [2] Xu et al., SQLNet: Generating Structured Queries From Natural Language Without Reinforcement Learning UPDATE: I thank the authors for addressing my concern about the comparison with baselines. However, I still feel that their approach relies on many dataset-specific design, and the technical novelty is incremental compared to prior work on semantic parsing. I would encourage the authors to evaluate their approach on other (semantic parsing) datasets as well, which may require them to remove some dataset-specific components in their current design.

Correctness: I think their approach is in the right direction of improving the performance on the CFQ benchmark, but not a general-purpose approach for compositional generalization in natural languages. The technique itself is sound, though some details are missing. The evaluation metric is unclear, and needs some more clarifications.

Clarity: The writing is generally fine, though some parts are unclear.

Relation to Prior Work: This paper lacks a discussion of existing work on leveraging syntactic structures for semantic parsing. In particular, they didn't discuss previous work on synthesizing SQL clauses as a set.

Reproducibility: Yes

Additional Feedback:

[Author Response · NeurIPS 2020]

We thank all reviewers for their time and efforts in reviewing our paper. We are grateful that you find this idea innovative and interesting. We will carefully revise this paper based on your constructive comments.

———————————————————— **Answers for Primary Issues** ————————————————————

**Can this method generalize to more domains?  (R3 & R4)** Yes, this method can be easily applied/extended to more domains.  It is not specifically designed for SPARQL. Different semantic parsing tasks use various semantic formalisms, including SPARQL, Prolog, $\lambda$-calculus, Abstract Meaning Representations (AMR) and Discourse Representation Theory (DRT). They have the abstract form of poset (i.e., directed acyclic graphs with partial permutation invariance), thus our method can generalize to them. Figure on the right shows AMR as an example. Our method abstracts them into the form of conjunctive logical queries (Equation 1), which are equivalent to Horn clause logic—an expressive formal semantics upon which Prolog is built. We chose to evaluate our method on CFQ dataset because: (1) SPARQL can be regarded as a representative of these formalisms; (2) currently, CFQ is the only realistic benchmark that comprehensively measure compositional generalization.

**Discussion of related work about NL-to-SQL. (R4)** Thanks for pointing this out. We will add discussion of them in the revised version. Seq2SQL (Zhong et al.) and SQLNet (Xu et al.) inspire us a lot. Our method solves some of their limitations, thus can generalize to more domains. (1) SQLNet is based on seq2set prediction (for column names) and slot filling (for OPs and values). This solution requires that each column should be unique in WHERE clauses, thus limiting its generalization. (2) Seq2SQL is based on RL, while we think this solution may suffer from sparse rewards in CFQ benchmark, as SPARQL queries are much longer and more complex than SQL queries in WikiSQL.

**It may be helpful to reduce abstractions and talk through a concrete example. (R2 & R3)**
Thanks for your kind reminder!  We will carefully revise Section 3 and 5.1 to improve the readability.  Here we show an example poset ("*Who directed and executive produced Night of Dark Shadows' sequel and directed and edited Corpse Bride*").  An intuitive explanation for Algorithm 1 is: we decode topology traversal paths in parallel; different paths are merged at variables (e.g., $x_1$) and entities (e.g., *Corpse_Bride*).

**The general take-away for compositional generalization in this paper. (R3)** In our method, both poset decoding and hierarchical mechanism are essential for compositional generalization: (1) poset decoding exploits the partial permutation invariance to prevent order bias (Line 36-49); (2) hierarchical mechanism disentangles high-level sketches from low-level semantic information (Line 138-139). They collaborate to improve compositional generalization. Section 6.3 analyzes the necessity of them in detail.

———————————————————— **Answers for Other Issues** ————————————————————

**The encoder part should also be equipped with compositional generalization. (R1)**
Thanks for your valuable suggestion! We will explore on how to enhance the encoder part in our future work.

**How query simplification in baselines is implemented. (R1)** This is a greedy algorithm that aims to group clauses into as few groups as possible. The idea behind this is similar to "combine like terms" in mathematics.

**Do we already know up front how many paths there are going to be, or is this set to some max value?  (R2)**
Neither. We preserve all paths with possibility >= 0.5 in the ESIM classifier.

**Explaining the step in Line 148 would be useful.  (R2)** Yes, here is a typo of $L_{S_Y}(v_i) = L_{S_Y}(v_j)$. An intuitive explanation for this step is: two vertexes should be merged if they share the same abstract token and the same neighbors.

**What is the evaluation metric?  (R4)** Thanks for pointing this out and we think it important.  We re-evaluate our baselines based on SPARQL semantic equivalence, and it improves the accuracy by only 1.6% on average.

**Impact of utilizing type constraint knowledge. (R4)** We further analyze syntactic correctness of SPARQL queries predicted by the LSTM baseline and find that only 0.17% of error cases are caused by syntactical mistakes.

**How to handle SELECT and FILTER? (R4)** (1) SELECT: our preliminary classifier (FastText) achieves 100% accuracy for predicting SELECT clause; (2) FILTER: we found that FILTER always co-occurrences with two specified predicates (marry and sibling), so we simply bind FILTERs to them.

**What is the benefit of learning a phrase table, instead of learning a neural network for prediction? (R4)**
We agree that a neural model would be more general, we will make a comparison in our revised version.

[Meta-Review · NeurIPS 2020]

The CFQ dataset designed for testing compositional generalization is really challenging. The presented results in this paper on CFQ are impressive. However, as pointed out by Reviewer #4, the proposed method relies on many dataset-specific design, and the technical novelty is incremental compared to prior work on semantic parsing. The work will be much more convincing if it can also be validated on another dataset.